# The Role of Adrenaline, Noradrenaline, and Cortisol in the Pathogenesis of the Analgesic Potency, Duration, and Neurotoxic Effect of Meperidine

**DOI:** 10.3390/medicina59101793

**Published:** 2023-10-09

**Authors:** Mehmet Yilmaz, Bahadir Suleyman, Renad Mammadov, Durdu Altuner, Seval Bulut, Halis Suleyman

**Affiliations:** 1Department of Orthopedics and Traumatology, 25 Aralık State Hospital, Gaziantep 27060, Turkey; 2Department of Pharmacology, Faculty of Medicine, Erzincan Binali Yildirim University, Erzincan 24100, Turkey; bahadirsuleyman@yandex.com (B.S.); renad_mamedov@hotmail.com (R.M.); durdualtuner@hotmail.com (D.A.); sevalbulut2010@hotmail.com (S.B.); halis.suleyman@gmail.com (H.S.)

**Keywords:** meperidine, adrenaline, noradrenaline, cortisol, analgesia, neurotoxicity

## Abstract

*Background and Objectives*: The purpose of the study was to investigate the role of adrenaline (ADR), noradrenaline (NDR), and cortisol in the pathogenesis of the analgesic potency, duration, and epilepsy-like toxic effect of meperidine. *Materials and Methods*: The experimental animals were separated into 11 groups of six rats. In the meperidine (MPD) and metyrosine + meperidine (MMPD) groups, paw pain thresholds were measured before and after the treatment between the first and sixth hours (one hour apart). In addition, ADR and NDR analyses were performed before and after the treatment, between the first and fourth hours (one hour apart). For the epilepsy experiment, caffeine, caffeine + meperidine, and caffeine + meperidine + metyrapone groups were created, and the treatment was applied for 1 day or 7 days. Groups were created in which caffeine was used at both 150 mg/kg and 300 mg/kg. Epileptic seizures were observed in epilepsy groups, latent periods were determined, and serum cortisol levels were measured. *Results*: In the MPD group, pain thresholds increased only at the first and second hours compared to pre-treatment, while ADR increased at the third hour, leading to a decrease in pain thresholds. In the MMPD group, the increase in paw pain thresholds at 1 and 6 h was accompanied by a decrease in ADR and NDR. In the caffeine (150 mg/kg) + meperidine group, 1-day treatment did not cause epileptic seizures, while seizures were observed and cortisol levels increased in the group in which treatment continued for 7 days. When cortisol levels were compared between the group in which caffeine (300 mg/kg) + meperidine + metyrapone was used for 7 days and the animals receiving caffeine (300 mg/kg) + metyrapone for 7 days, it was found that cortisol levels decreased and the latent period decreased. *Conclusions*: The current study showed that if serum ADR and cortisol levels are kept at normal levels, a longer-lasting and stronger analgesic effect can be achieved with meperidine, and epileptic seizures can be prevented.

## 1. Introduction

Meperidine is a synthetic analgesic drug, which is the hydrochloride salt of an opioid classified as phenylpiperidine [1]. Phenylpiperidines are a chemical class of drugs in which the phenyl moiety is directly bonded to piperidine. These agents have an important role in many areas of medicine, including anesthesia and pain medicine [2]. Meperidine was the first synthetic opioid used in humans [3]. In the following years, derivatives such as fenoperidine and feneridine were developed [4]. Meperidine was the most commonly used opiate in clinical anesthesia for over 20 years until the development of fentanyl [5]. The action of meperidine is mediated through the binding of opioid receptors in the heart, lungs, blood vessels, intestines, and the central nervous system. In addition, meperidine is anticholinergic in nature [1]. The delta receptor may be involved in opioid addiction, whereas the mu and kappa receptors are thought to be predominantly involved in analgesia [6]. Currently, meperidine is used in the treatment of moderate and severe chronic pain in both oral and parenteral forms [7,8]. Furthermore, meperidine has been found to be effective in treating pain following major orthopedic surgery and after total hip or total knee replacements [9]. Meperidine acts as an agonist at mu-opioid receptors and follows a similar mechanism of action to morphine [10]. The analgesic effect of meperidine, however, is weaker than that of morphine, and its duration of action is shorter than that of morphine [10]. Therefore, meperidine is insufficient to control pain [11]. To determine the reason for the short duration of the analgesic effect of meperidine, previous studies have been reviewed and it has been found that meperidine significantly increases plasma levels of epinephrine (ADR) and norepinephrine (NDR) [12]. Based on the results of a previous study, it has been found that rats with reduced serum ADR levels experience longer anesthetic effects than healthy rats and it was reported that ketamine increased the endogenous catecholamine level in a dose-dependent manner [13]. In this study by Aksoy et al., it was emphasized that ketamine should suppress the amount of endogenous ADR to a certain level to prolong the duration of anesthesia; it has been stated that if endogenous adrenaline is suppressed, ketamine can provide adequate anesthesia even at a 2-fold lower dose [13]. According to another study, the analgesic effects of nonsteroidal anti-inflammatory drugs were seen to be analgesic effects related to endogenous adrenaline and cortisol levels [14]. Based on this information obtained from the literature, it appears that the relatively weak analgesia of meperidine, in addition to the short duration of its action, may be attributed to the higher level of serum ADR.

One of the disadvantages of meperidine is the occurrence of undesirable side effects similar to those of morphine [5]. Meperidine has the serious disadvantage of being neurotoxic [15]. The neurotoxic effects of meperidine, including the development of epileptic seizures, have not yet been explained by opioid pathways [5]. In a study by Hacımüftüoğlu et al., it was suggested that a rise in endogenous cortisol contributed to epileptic seizures [16]. It has been reported in a recent study that the serum cortisol level increases following the administration of opioids [17]. In another study, cortisol levels were observed to increase in animals treated with meperidine, although the increase in cortisol could not be attributed to meperidine [18]. Information obtained from the literature indicates that the epilepsy-like neurotoxic effect of meperidine may be associated with increased cortisol levels. Furthermore, maintaining normal levels of blood ADR and cortisol may enhance the analgesic effect and duration of meperidine and prevent epilepsy-like seizures. The purpose of this study was to investigate the role of ADR, NDR, and cortisol in the pathogenesis of the analgesic potency, duration, and epilepsy-like toxic effects of meperidine.

## 2. Material and Methods

### 2.1. Animals

A total of 66 5–6-month-old albino male Wistar rats, each weighing between 345 and 360 g, were used in this experiment. All the experimental animals were obtained from Erzincan Binali Yıldırım University Experimental Animals Application and Research Center. To acclimatize the animals to the experimental environment, each group was housed in a separate cage at normal room temperature (22 °C) and fed ad libitum.

### 2.2. Chemical Substances

The thiopental sodium (500 mg flakon) to be used in the experiment was procured from I.E. Ulagay (Istanbul, Turkey), meperidine (100 mg/2 cc ampoule) was procured from Liba Laboratories (Istanbul, Turkey), Metyrapone (Metopirone 250 mg capsule) from Alliance Pharmaceutical Ltd. (Wiltshire, UK), Metyrosin (Demser 250 mg capsule) from Merck (Kenilworth, NJ, USA), and and Caffeine (5 g powder) from Sigma-Aldrich (Darmstadt, Germany). Meperidine was administered intramuscularly (im), approximately 0.15 cc per rat, by calculating the dose per weight. Metirozine and metyrapone were powdered for oral use. A total of 250 mg/5 mL solutions were prepared by adding distilled water. It was administered orally at 1 cc per rat. Thiopental sodium was diluted with physiological saline to prepare a 500 mg/5 cc solution. It was administered intraperitoneally (ip), 0.2 cc per rat. Caffeine, was diluted with physiological saline to prepare 500 mg/1 cc solution. It was administered ip, 0.1 (150 mg/kg) or 0.2 (300 mg/kg) cc per rat.

### 2.3. Experimental Groups

As seen in Table 1, the experimental animals were categorized into 11 groups, Healthy group (HG), Meperidine (MPD), Metyrosin + Meperidine (MMPD), Caffeine (C-150), Caffeine (C-300), Meperidine + Caffeine (MC-150), Meperidine + Caffeine (MC-300), Meperidine + Caffeine (MCL-150), Meperidine + Caffeine (MCL-300), Metyrapone + Meperidine + Caffeine (MMC-150), and Metyrapone + Meperidine + Caffeine (MMC-300).

### 2.4. Experimental Procedure

#### 2.4.1. Pain Threshold Test

It has been stated in the literature that meperidine causes an increase in catecholamine levels [12]. Studies have shown that changes in serum ADR levels affect the duration of action of ketamine and the analgesic power of NSAIDs [13,14]. Therefore, paw pain threshold measurements and determination of serum ADR and NADR levels were included in the study. To evaluate the analgesic effect of the treatments in the HG (*n* = 6), MPD (*n* = 6), and MMPD (*n* = 6) groups, paw pain thresholds were measured with the Basile Algesimeter once before and once after the treatments at one-hour intervals (1–6 h). Time-dependent analgesic activity was calculated using paw pain threshold values. The following formula was used to determine analgesic activity: analgesic activity = 100 − (pre-treatment paw pain threshold/post-treatment paw pain threshold × 100). For example, the percentage of analgesic activity of MPD in the first hour = 100 − (30.33/46.17 × 100) = 34.05%. After the initial paw pain threshold measurements, the MMPD group was then given metyrosin (150 mg/kg), and the MPD group was given the same volume of distilled water by oral gavage. Metyrosin is a tyrosine hydroxylase inhibitor that blocks the conversion of tyrosine to dopa, the rate-limiting step in catecholamine synthesis. Metyrosin use reduces ADR and NDR biosynthesis [19]. In this study, we aimed to suppress the increase in ADR and NDR levels with meperidine in the MMPD group with the use of metyrosin. In the MMPD and MPD groups, meperidine was injected im at a dose of 20 mg/kg one hour after the administration of metyrosin and distilled water. To measure serum levels of ADRs and NDRs, blood samples were taken from the tail vein before meperidine injection and 1 and 4 h (1 h apart) after injection. At the end of the processes, the animals were euthanized by ip administration of thiopental sodium (50 mg/kg). 

#### 2.4.2. Epilepsy Test

In this study, caffeine was used to create an epileptic model. Caffeine is a frequently preferred agent to create epileptic models in animals by giving different doses, including high doses [20]. It is mentioned in the literature that meperidine has proconvulsant activity [21]. In the caffeine-induced epileptic model, the effect of meperidine on seizures was tried to be evaluated by creating caffeine and caffeine + meperidine applied groups. To determine the epileptic and subepileptic doses of caffeine for this stage of the study, caffeine was injected ip at a dose of 150 mg/kg in the C-150 group (*n* = 6) and at a dose of 300 mg/kg in the C-300 group (*n* = 6). The HG group received the same volume of ip distilled water. In the MC-150 group (*n* = 6), meperidine was administered im twice at a dose of 20 mg/kg (6 h intervals). In this group, 150 mg/kg caffeine was administered intraperitoneally one hour following the last meperidine injection. 

Meperidine 20 mg/kg was administered twice (6 h intervals) to the MC-300 group (*n* = 6). A dose of 300 mg/kg of caffeine was administered ip in this group one hour following the last dose of meperidine.

In the MCL-150 group (*n* = 6), meperidine was administered twice daily (6 h intervals) for 7 days at a dose of 20 mg/kg im. At 1 h after the last injection of meperidine on day 7, 150 mg/kg of caffeine was administered ip to this group.

The MCL-300 group (*n* = 6) received meperidine im twice daily (6 h intervals) for 7 days at a dose of 20 mg/kg. One hour after the last dose of meperidine injection, 300 mg/kg caffeine was injected ip.

In the literature, an increase in cortisol levels has been associated with epileptic seizures [16]. In this study, metyrapone was used in the MMC-150 and MMC-300 groups to evaluate the role of cortisol in the pathogenesis of epileptic seizures. Metyrapone inhibits the conversion of 11-deoxycortisol to corticosterone by inhibiting 11-β hydroxylase in rats [22]. In the MMC-150 group (*n* = 6), 150 mg/kg of metyrapone was administered orally once a day to the stomach. One hour after metyrapone was administered, meperidine at a dose of 20 mg/kg was injected im twice a day (6 h intervals). This procedure was repeated for seven consecutive days. At 1 h after the last dose of meperidine was administered, 150 mg/kg of caffeine was injected ip.

In the MMC-300 group (*n* = 6), 150 mg/kg of metyrapone was administered once daily orally to the stomach. At 1 h after the administration of metyrapone at a dose of 20 mg/kg twice daily (6 h intervals) meperidine im was administered. This procedure was repeated for 7 days. One hour after the last dose of meperidine was administered, 300 mg/kg of caffeine was injected ip. 

Immediately after the caffeine injection, the rats in each group were placed in a plexiglass box (30 × 30 × 40 cm), and the time to the onset of tonic–clonic contractions (latent period) was measured with a stopwatch in seconds. Additionally, levels of cortisone (corticosterone in rats) levels were analyzed in blood samples taken from all epilepsy groups. For animals experiencing seizures, blood was taken during the contraction period. At the end of the processes, the animals were euthanized by ip administration of thiopental sodium (50 mg/kg).

### 2.5. Analysis of Serum Corticosterone Levels 

Blood samples were collected from the rats into EDTA vacuum tubes to determine the corticosterone levels. Samples were centrifuged at 3500 rpm for 10 min, then frozen and kept at −80 °C until measurement day. The plasma was separated and extracted with 5 mL of ethyl acetate (betamethasone as the internal standard), and then the extract was washed with sodium hydroxide (0.1 M) and water. After evaporation of the ethyl acetate, the residue was dissolved in the mobile phase (acetonitrile-water-acetic acid-TEA, 22:78:0.1:0.03, *v*/*v*) and injected into an isocratic HPLC consisting of a 10 cm C18 column and UV detector at 254 nm. The plasma corticosterone concentration was measured with an isocratic system using an HPLC pump (model Hewlett Packard Agilent 1100) (flow rate: 1 mL/min; injection volume: 150 μL). Pure corticosterone (Sigma, St. Louis, MO, USA) was provided, dissolved in ethyl acetate. The samples were applied directly and compared with standard pure corticosterone.

### 2.6. Analysis of Serum Adrenaline and Noradrenaline Levels

For the determination of adrenaline and noradrenaline levels, blood samples were placed in vacuum tubes containing ethylenediamine tetra-acetic acid (EDTA) and centrifuged at 3500 rpm for 5 min. A high-performance liquid chromatography (HPLC) pump was then used to measure adrenaline and noradrenaline concentrations in plasma (Hewlett Packard Agilent 1100; Hewlett Packard Enterprise, Spring, IL, USA; flow rate: 1 mL/min; injection volume). A reagent kit was used for HPLC analysis of catecholamines in plasma serum (Chromsystems, Munich, Germany).

### 2.7. Statistical Analyses

The study data were analyzed statistically using “IBM SPSS Statistics vn. 22.0 software”. For repeated measures, the data were analyzed using a (dependent) Repeated Measures Analysis of Variance (ANOVA), and for independent data, One-way Analysis of Variance (ANOVA) was performed. The Bonferroni test was used for pairwise comparisons. The results obtained from the experiments were expressed as “mean value ± standard error” (X ± SEM). A value of *p* < 0.05 was considered statistically significant.

## 3. Results

### 3.1. Pain Test Results

As presented in Table 2 and Figure 1, the paw pain threshold values of the MPD group increased significantly at the first (*p* = 0.011) and second (*p* = 0.005) hours compared to pre-treatment values. The pain threshold levels at the third and sixth hours were found to be close to the levels before meperidine administration (*p* > 0.05), and the difference was not statistically significant. The paw and pain threshold measurement values of the MMPD group were found to be higher than the values before treatment at the first and sixth hours (*p* < 0.05). 

In the MPD group, the analgesic activity of meperidine decreased after the third hour (Table 2, Figure 1). In the metyrosin + meperidine combination group, analgesic activity persisted for all time periods measured following the treatment.

### 3.2. Epilepsy Test and Serum Corticosterone Analysis Results

As seen in Table 3, no seizures were observed in the HG group. Serum cortisone levels were 4.31 ± 0.07. In the C-150 group, no tonic or tonic–clonic contraction was observed in any of the animals administered 150 mg/kg dose of caffeine. The mean blood cortisol level in this group was 4.38 ± 1.7 ng/mL. In all animals in the C-300 group treated with 300 mg/kg caffeine, severe tonic and tonic–clonic contractions were seen to occur within an average of 152.00 ± 2.54 s, and all animals died as a result. The mean blood cortisol level in this group was found to be 4.27 ± 0.20 ng/mL. These findings confirmed that the subepileptic dose of caffeine was 150 mg/kg, and the epileptic dose was 300 mg/kg. The difference between blood cortisol levels in C-150, C-300, and HG was not significant (*p* > 0.05).

Although meperidine was administered twice a day at 6 h intervals before caffeine administration, no epilepsy-like seizures were observed in any of the rats. These findings show that meperidine alone does not cause epileptic seizures. Caffeine at 150 mg/kg dose did not cause convulsions in any of the animals in the MC-150 group. The mean blood cortisol level in this group was found to be 4.23 ± 0.11 ng/mL. Contractions occurred at an average of 149.17 ± 2.98 s in the MC-300 group treated with 300 mg/kg of caffeine, and all the animals died. This group had a mean blood cortisol level of 4.08 ± 0.14 ng/mL. The difference between blood cortisol levels for MC-150, MC-300, and HG groups was not significant (*p* > 0.05).

No epilepsy-like seizures were observed in any of the animals administered meperidine twice a day at 6 h intervals for 7 days before caffeine. These findings indicated that meperidine alone does not cause epileptic seizures. Of all the animals in the MCL-150 group, caffeine at a dose of 150 mg/kg caused contractions in an average of 198.33 ± 3.20 s, but no animals died from seizures. A mean cortisol level of 7.07 ± 0.17 ng/mL was measured in this group. The animals in the MCL-300 group, which received 300 mg/kg caffeine, all had contractions in an average of 79.33 ± 2.35 s, and all of the animals in this group died. The blood cortisol level was determined to have increased to 7.27 ± 0.20 ng/mL.

A dose of 150 mg/kg of caffeine did not cause contractions or death in any of the animals in the MMC-150 group. A blood cortisol level of 4.05 ± 0.10 ng/mL was determined for the animals in this group. With a dose of 300 mg/kg of caffeine, the animals in the MMC-300 group had contractions in an average of 157.17 ± 2.12 s, and every animal in this group died. In the MMC-300 group, the blood cortisol level was found to be 4.20 ± 0.21 ng/mL. 

### 3.3. Serum ADR and NDR Analysis Results

According to Table 2 and Figure 2, the serum ADR level in the MPD group was 1088.33 ± 16.34 ng/L prior to meperidine administration. At 1, 2, 3, and 4 h following meperidine administration, serum ADR levels were determined to be 1091.17 ± 6.90 ng/L, 1100.67 ± 4.13 ng/L, 1425.17 ± 7.91 ng/L, and 1435.50 ± 5.12 ng/L, respectively. Compared to the pre-treatment period, adrenaline levels in the MPD group did not differ at the first and second hours (*p* = 1000), but there was a significant difference at the third and fourth hours (*p* < 0.001). The serum NDR level in the MPD group was 513.67–5.45 ng/L before meperidine administration, and 519.00 ± 18.87 ng/L, 508.83 ± 37.78 ng/L, 507.83 ± 37.62 ng/L, and 513.00 ± 35.81 ng/L at the first, second, third, and fourth hours following meperidine administration, respectively. At all the measured time points, the noradrenaline levels were similar in the MPD group (*p* = 1.000).

An ADR level of 1096.17 ± 12.67 ng/L was determined prior to the application of the metyrosin and meperidine combination to the MMPD group of animals. The serum ADR levels following the administration of the metyrosin + meperidine combination were 701.33 ± 10.63 ng/L, 697.00 ± 9.60 ng/L, 691.00 ± 7.18 ng/L, 791.00 ± 5.97 ng/L at the first, second, third, and fourth hours, respectively. The serum NDR level in the MMPD group was found to be 526 ng/L before the drugs were administered, and the levels were determined to be 525.83 ± 3.83 ng/L, 208.00 ± 5.68 ng/L, 225.33 ± 4.86 ng/L, 232.67 ± 3.61 ng/L and 252.67 ± 4.21 ng/L at the first, second, third, and fourth hours after administration, respectively. As a result of the treatment, the levels of adrenaline and noradrenaline decreased at all measured time points (*p* < 0.001) (Table 2, Figure 2). 

## 4. Discussion

The main purpose of this study was to achieve strong and long-lasting analgesia with meperidine and to alleviate the neurotoxic effects of meperidine that prevent its widespread use. Based on the information obtained from the literature [12,13,14], it was thought that the weak and short-term analgesic effects of meperidine were possibly caused by an increase in serum ADR levels. According to the results of this study, meperidine alone produced significant analgesia in rats for two hours, during which time serum levels of ADR and NDR were not significantly affected by meperidine. However, meperidine significantly increased only serum ADR levels at 3 and 4 h. Experimental results and information from the literature suggest that the weak and short-term analgesic effect of meperidine is due to high serum ADR levels. In a previous study, the duration of the anesthetic effect of ketamine was associated with serum ADR levels [13]. Another study demonstrated the role of endogenous ADR in the pathogenesis of the analgesic effects of nonsteroidal anti-inflammatory drugs was revealed [14]. In the current study, the analgesic effect of meperidine was also evaluated in rats with suppressed ADR production by methyrosine. Metyrosin inhibits catecholamine synthesis by inhibiting tyrosine hydroxylase, resulting in a 35–80% reduction in catecholamine levels [23]. The results of this study indicated that meperidine produced stronger and longer analgesia in animals with ADR levels reduced by metyrosin. As a result of these findings, it was suggested that the combination of meperidine and metyrosin inhibited pain through different pathways. A previous study supporting this idea reported that β2-adrenergic receptors played a role in analgesia [14]. In the current study, endogenous cortisol was able to potentiate the analgesic effect of meperidine by stimulating β2-adrenergic receptors in animals with ADR levels suppressed with metyrosin. 

Meperidine has the serious disadvantage of being neurotoxic [15]. However, opioid pathways cannot account for the mechanisms of neurotoxic effects such as meperidine-induced epileptic seizures [5]. According to a recent study, opioids increase serum cortisol levels [17]. In the literature, it has been suggested that an increase in endogenous cortisol may play an important role in the pathogenesis of epilepsy [16]. As can be understood from our experimental results, the proepileptic activity of caffeine did not change in the animals administered meperidine for a short period of time in which blood cortisol level was not significantly affected. According to the literature, low-dose corticosterone treatment did not affect epilepsy severity [16]. However, cortisol levels increased significantly in animals treated with meperidine for 7 days. However, cortisol levels increased significantly in animals treated with meperidine for 7 days. At a dose of 300 mg/kg, caffeine caused more severe contractions in a shorter time in the group treated with meperidine for a long period of time in which cortisol levels increased. Caffeine caused contractions even at subepileptic doses (150 mg/kg). These findings pointed to the role of cortisol in the pathogenesis of meperidine’s proepileptic effect. Several clinical studies have demonstrated a relationship between plasma cortisol and neurological disorders [24]. As can be seen from our experimental results, all the animals in the meperidine group with the highest cortisol levels died. Corticosterone levels or chronic corticosteroid treatment have been reported to induce epileptogenesis [25,26]. The increase in corticosterone levels increased the degree of epileptogenesis by altering the functions of hippocampal cells [27]. Metyrapone was also used in this study to prevent cortisol increases related to meperidine use. The results of this study indicated that metyrapone suppressed the increase in cortisol levels associated with meperidine use. In meperidine group animals in which cortisol increase was suppressed by metyrapone, caffeine caused convulsions at proepileptic dose but not at subepileptic dose. It is well known that metyrapone has been used to treat hypercortisolism in patients suffering from Cushing’s syndrome [28]. Metyrapone is an inhibitor of 11β-hydroxylase that contributes to the synthesis of cortisol [22]. Therefore, it has been used to suppress cortisol production and lower its level [29]. According to the findings of the study and the literature, it can be said that the significant increase in serum cortisol level is responsible for the shortening of the latent period in the group administered meperidine for a long time.

The limitations of this study were that the analgesic effect of meperidine was measured for six hours at one-hour intervals. Since it was not possible to obtain sufficient blood from the rats, it was only possible to measure the serum ADR level before the injection of meperidine as well as at the first and fourth hour after injection. Further, more detailed studies are required to clarify the action mechanism of meperidine in terms of its analgesic and neurotoxic effects. 

## 5. Conclusions

As has been stated, meperidine’s short-term analgesic effect was due to the increase in serum ADR levels. Furthermore, short-term use of meperidine did not change serum cortisol levels, and 150 mg of caffeine did not cause epilepsy-like contractions. In contrast, long-term use of meperidine, even at low doses, resulted in increased cortisol levels and contractions similar to those associated with tonic–clonic epilepsy. In our study, the cortisol levels in the blood samples of animals administered 150 and 300 mg/kg doses of caffeine alone were found to be close to those of the group administered meperidine + caffeine and were statistically insignificant. In addition, cortisol level was found to be significantly higher in the group administered meperidine twice a day for 7 days. The epilepsy model in this study was induced with a single dose of caffeine in animals administered short and long-term meperidine. The fact that the cortisol level in the animals administered a single dose of caffeine was close to that of the healthy group indicates that the blood cortisol level increases with meperidine. According to these experimental results and the information obtained from the literature, it may be advantageous to keep blood ADR and cortisol levels under control at normal levels to increase the analgesic effect and duration, and for meperidine, to prevent epileptic seizures. 

## Figures and Tables

**Figure 1 medicina-59-01793-f001:**
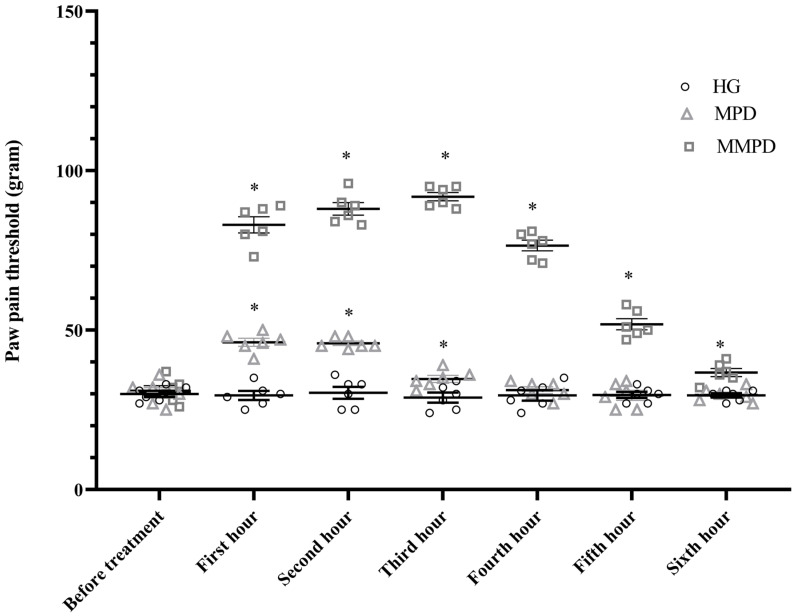
Time-dependent variation of paw pain threshold values of the MPD and MMPD groups. *; *p* < 0.05 vs. before treatment, HG; healthy group, MPD; meperidin group, MMPD; metyrosin + meperidin group.

**Figure 2 medicina-59-01793-f002:**
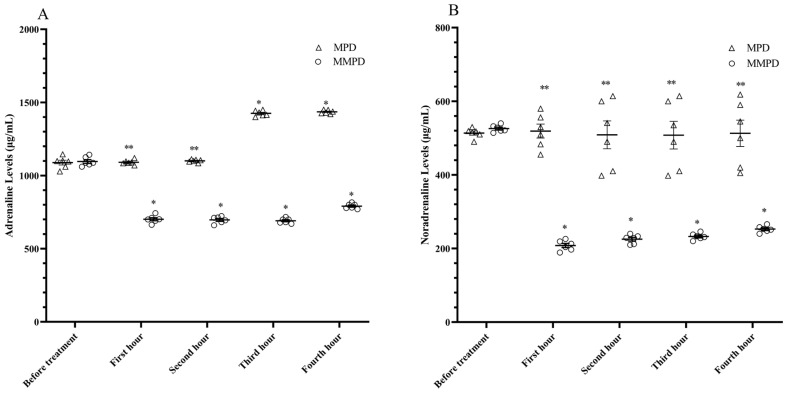
(**A**,**B**). Time-dependent variation of adrenaline (**A**) and noradrenaline (**B**) levels obtained from serum of MPD and MMPD groups. *; *p* < 0.05 vs. before treatment, **; *p* > 0.05 vs. before treatment, MPD; meperidin group, MMPD; metyrosin + meperidin group.

**Table 1 medicina-59-01793-t001:** Experimental groups.

Groups	Drugs	Dose	Routes of Administration	Treatment Duration
HG	distilled water	1 mL	im	2 doses (6 h apart), 1 day
			ip	1 dose, 1 day
MPD	meperidine	20 mg/kg	im	2 doses (6 h apart), 1 day
MMPD	meperidine	20 mg/kg	im	2 doses (6 h apart), 1 day
	metyrosin	150 mg/kg	orally	1 dose, 1 day
C-150	caffeine	150 mg/kg	ip	1 dose, 1 day
C-300	caffeine	300 mg/kg	ip	1 dose, 1 day
MC-150	meperidine	20 mg/kg	im	2 doses (6 h apart), 1 day
	caffeine	150 mg/kg	ip	1 dose, 1 day
MC-300	meperidine	20 mg/kg	im	2 doses (6 h apart), 1 day
	caffeine	300 mg/kg	ip	1 dose, 1 day
MCL-150	meperidine	20 mg/kg	im	2 doses (6 h apart), 7 days
	caffeine	150 mg/kg	ip	1 dose, 7th day
MCL-300	meperidine	20 mg/kg	im	2 doses (6 h apart), 7 days
	caffeine	300 mg/kg	ip	1 dose, 7th day
MMC-150	meperidine	20 mg/kg	im	2 doses (6 h apart), 7 days
	caffeine	150 mg/kg	ip	1 dose, 7th day
	metyrapone	150 mg/kg	orally	1 dose, 7 days
MMC-300	meperidine	20 mg/kg	im	2 doses (6 h apart), 7 days
	caffeine	300 mg/kg	ip	1 dose, 7th day
	metyrapone	150 mg/kg	orally	1 dose, 7 days

ip: intraperitoneally, im: intramuscularly.

**Table 2 medicina-59-01793-t002:** Analysis results of paw pain threshold and serum adrenaline and noradrenaline data obtained from rats in tHG, MPD and MMPD groups.

Parameters			HG	MPD	MMPD
Paw pain threshold (gram)	Before treatments		30.00 ± 0.97	30.33 ± 1.61	31.00 ± 1.59
	After treatments	First hour	29.50 ± 1.41 **	46.17 ± 1.25 *	83.00 ± 2.52 *
		Second hour	30.33 ± 1.86 **	45.83 ± 0.70 *	88.00 ± 1.95 *
		Third hour	28.83 ± 1.60 **	34.67 ± 1.12 **	91.83 ± 1.30 *
		Fourth hour	29.50 ± 1.61 **	31.17 ± 1.08 **	76.50 ± 1.69 *
		Fifth hour	29.67 ± 0.96 **	29.50 ± 1.59 **	51.83 ± 1.74 *
		Sixth hour	29.50 ± 0.67 **	29.67 ± 0.88 **	36.67 ± 1.28 *
*p* values			0.480	<0.001	<0.001
Analgesic activity (%)		First hour	−2.15 ± 2.39 ^a^	34.05 ± 3.89 ^a^	62.54 ± 1.97 ^ab^
		Second hour	0.13 ± 3.34 ^a^	33.78 ± 3.45 ^a^	64.77 ± 1.65 ^a^
		Third hour	−4.78 ± 2.82 ^a^	12.54 ± 3.36 ^b^	66.25 ± 1.61 ^a^
		Fourth hour	−2.42 ± 3.10 ^a^	2.93 ± 2.28 ^b^	59.47 ± 1.92 ^b^
		Fifth hour	−1.16 ± 1.13 ^a^	−2.92 ± 1.62 ^b^	39.97 ± 3.38 ^c^
		Sixth hour	−1.69 ± 2.22 ^a^	−2.19 ± 4.34 ^b^	15.62 ± 2.10 ^d^
*p* values			0.488	<0.001	<0.001
Adrenaline	Before treatments			1088.33 ± 16.34	1096.17 ± 12.67
	After treatments	First hour		1091.17 ± 6.90 **	701.33 ± 10.63 *
		Second hour		1100.67 ± 4.13 **	697.00 ± 9.60 *
		Third hour		1425.17 ± 7.91 *	691.00 ± 7.18 *
		Fourth hour		1435.50 ± 5.12 *	791.00 ± 5.97 *
*p* values				<0.001	<0.001
Noradrenaline	Before treatments			513.67 ± 5.45	525.83 ± 3.83
	After treatments	First hour		519.00 ± 18.87 **	208.00 ± 5.68 *
		Second hour		508.83 ± 37.78 **	225.33 ± 4.86 *
		Third hour		507.83 ± 37.62 **	232.67 ± 3.61 *
		Fourth hour		513.00 ± 35.81 **	252.67 ± 4.21 *
*p* values				0.886	<0.001

*; *p* < 0.05 vs. before treatment, **; *p* > 0.05 vs. before treatment. For analgesic activities of the same group calculated at different times; *p* > 0.05 for different letters, *p* > 0.05 for the same letters. HG; healthy group, MPD; meperidin group, MMPD; metyrosin + meperidin group. Statistical analysis was performed with repeated measures ANOVA, followed by the Bonferroni test.

**Table 3 medicina-59-01793-t003:** Analysis results of epileptic seizure and serum corticosterone data obtained from experimental groups.

Groups	Latent Period (s)	Number of Rats with Epileptic Seizures	Number of Dead Animals	Corticosterone Levels
HG	-	-	-	4.31 ± 0.07 *
C-150	-	0	0	4.38 ± 1.7 *
C-300	152.00 ± 2.54 *	6	6	4.27 ± 0.20 *
MC-150	-	0	0	4.23 ± 0.11 *
MC-300	149.17 ± 2.98 *	6	6	4.08 ± 0.14 *
MCL-150	198.33 ± 3.20 **	6	0	7.07 ± 0.17 **
MCL-300	79.33 ± 2.35 ^#^	6	6	7.27 ± 0.20 **
MMC-150	-	0	0	4.05 ± 0.10 *
MMC-300	157.17 ± 2.12 *	6	6	4.20 ± 0.21 *
*p* values	<0.001			<0.001

*p* < 0.001, for different symbols in the same column; *p* > 0.05, for same symbols in the same column. HG; Healthy group, C-150; 150 mg/kg caffeine administered group, C-300; 300 mg/kg caffeine administered group, MC-150; 1 day 2 × 20 mg/kg meperidine + 150 mg/kg caffeine (1 day), MC-300; 1 day 2 × 20 mg/kg meperidine + 300 mg/kg caffeine, MCL-150; 7 days 2 × 20 mg/kg meperidine +150 mg/kg caffeine, MCL-300; 7 days 2 × 20 mg/kg meperidine + 300 mg/kg caffeine, MMC-150; 7 days 150 mg/kg metyrapone + 2 × 20 mg/kg meperidine + 150 mg/kg caffeine, MMC-300; 7 days 150 mg/kg metyrapone + 2 × 20 mg/kg meperidine + 300 mg/kg caffeine. Statistical analysis was performed with One-way ANOVA, followed by the Bonferroni test.

## Data Availability

It can be obtained from the corresponding author upon request.

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
