# Peer review of "The Role of Adrenaline, Noradrenaline, and Cortisol in the Pathogenesis of the Analgesic Potency, Duration, and Neurotoxic Effect of Meperidine"

_medicina, 2023, doi:10.3390/medicina59101793_

Round 1
Reviewer 1 Report
Remarks to the author:
In this research article, the authors explored the impact of adrenaline, noradrenaline, and cortisol on meperidine's analgesic efficacy, duration, and propensity to induce epilepsy-like effects in Albino Wistar rats. They administered varying doses of caffeine to induce epilepsy in the animals. Their findings revealed that meperidine-treated animals exhibited elevated pain thresholds for a duration of 2 hours, which gradually decreased as adrenaline levels increased, resulting in reduced pain thresholds. Additionally, they noted that short-term treatment did not lead to epileptic seizures. However, prolonged treatment raised cortisol levels and subsequently induced seizures in these rats. Consequently, the authors concluded that maintaining normal levels of adrenaline and cortisol may enhance the analgesic effectiveness and duration of meperidine while also preventing the occurrence of epileptic seizures.
Specific comments:
i) Abstract: the methods section does not adequately emphasize the specific measurements considered in this study. Mentioning all 10 study groups in the abstract makes it unnecessarily complex.
ii) Authors should provide justification for the utilization of various drugs such as metyrosin, caffeine, and meperidine when describing the different groups used in the study. This rationale should be presented within the main body of the paper, ideally in the section discussing the experimental design, to help readers comprehend the purpose behind employing distinct groups of rats treated with these drugs.
iii) The authors' choice of using caffeine to induce epilepsy instead of directly assessing the impact of meperidine on epilepsy is not clearly explained.
iv) The authors should elucidate the criteria they employed for selecting the two different caffeine doses in the animals, particularly considering that the 300 mg/kg dose exhibited lethal effects in a majority of the groups.
v) It is essential to distinguish the effects of cortisol levels in response to meperidine from the effects of caffeine on cortisol levels. Further clarification is required in this regard.
vi) The study lacks results for a normal, healthy control group, which would have provided a baseline for the physiological levels of the various parameters measured in the study.
vii) Table 2 presents a discrepancy with the preceding text, where it's stated that all rats in the MMC-300 group died. However, the table indicates that no animals in that group perished. This inconsistency needs clarification.
viii) In the conclusion, it is mentioned that "150 mg did not cause epilepsy-like contractions." It is essential to specify which drug in the 150 mg dose is being referred to in this context.
ix) The authors should thoroughly review the entire manuscript for clarity, grammar, and typos.
The authors should thoroughly review the entire manuscript for clarity, grammar, and typos.
Author Response
Comments 1: Abstract: the methods section does not adequately emphasize the specific measurements considered in this study. Mentioning all 10 study groups in the abstract makes it unnecessarily complex.
Response 1: The methods section of the abstract has been rewritten. Page: 2, Line: 40-48
Comments 2: Authors should provide justification for the utilization of various drugs such as metyrosin, caffeine, and meperidine when describing the different groups used in the study. This rationale should be presented within the main body of the paper, ideally in the section discussing the experimental design, to help readers comprehend the purpose behind employing distinct groups of rats treated with these drugs.
Response 2: Justifications for the use of drugs were added to the material and method section. Page, line. Page: 5, Line: 168-172, 182-186; Page: 6 Line:198-202
Comments 3) The authors' choice of using caffeine to induce epilepsy instead of directly assessing the impact of meperidine on epilepsy is not clearly explained.
Response 3: Additions were made to the findings section. ‘Although meperidine was administered twice a day at 6-hour intervals before caffeine administration, no epilepsy-like seizures were observed in all animals. These findings show that meperidine alone does not cause epileptic seizures. Caffeine at 150 mg/kg dose did not cause convulsions in any of the animals in the MC-150 group. The difference between blood cortisol levels for MC-150, MC-300 and HG groups was not significant (p>0.05). None of the animals administered meperidine twice a day at 6-hour intervals for 7 days before caffeine administration did not show epilepsy-like seizures. These findings indicate that meperidine alone does not cause epileptic seizures.’
Comments 4) The authors should elucidate the criteria they employed for selecting the two different caffeine doses in the animals, particularly considering that the 300 mg/kg dose exhibited lethal effects in a majority of the groups.
Response 4: Additions were made to the materials and methods (epilepsy test) section to clarify the selection of caffeine doses. Page: 6, Line: 202-203
Comments 5) It is essential to distinguish the effects of cortisol levels in response to meperidine from the effects of caffeine on cortisol levels. Further clarification is required in this regard.
Response 5: The explanation regarding this issue was added to the conclusion section. ‘In our study, the cortisol levels in the blood samples of animals administered 150 and 300 mg/kg doses of caffeine alone were found to be close to those of the group administered meperidine+caffeine and were statistically insignificant. In addition, cortisol level was found to be significantly higher in the group administered meperidine twice a day for 7 days. The epilepsy model in this study was induced with a single dose of caffeine in animals administered short and long-term meperidine. The fact that the cortisol level in the animals administered a single dose of caffeine was close to that of the healthy group indicates that the blood cortisol level increases with meperidine’
Comments 6) The study lacks results for a normal, healthy control group, which would have provided a baseline for the physiological levels of the various parameters measured in the study.
Response 6: Healthy group data was added to the study.
Comments 7) Table 2 presents a discrepancy with the preceding text, where it's stated that all rats in the MMC-300 group died. However, the table indicates that no animals in that group perished. This inconsistency needs clarification.
Response 7: Corrected misspelling (6 instead of 0).
Comments 8) In the conclusion, it is mentioned that "150 mg did not cause epilepsy-like contractions." It is essential to specify which drug in the 150 mg dose is being referred to in this context.
Response 8: The deficiency was corrected as "150 mg caffeine".
Comments 9) The authors should thoroughly review the entire manuscript for clarity, grammar, and typos.
Response 9: The text has been edited for language. Page: 15, Line: 493

Reviewer 2 Report
The manuscript entitled "The Role of Adrenaline, Noradrenaline, and Cortisol in the Pathogenesis of the Analgesic Potency, Duration, and Neurotoxic Effect of Meperidine" by Yilmaz et al described the role of adrenaline (ADR), noradrenaline (NDR) and cortisol in the pathogenesis of meperidine's analgesic potency, duration and epilepsy-like toxic effect. The experimental animals were categorized under10 groups and treated with different conc. of meperidine (MPD), metyrosin with meperidine (MMPD), Caffeine (C-150), Caffeine (C-300), 20 mg/kg meperidine (2 doses every 6 hours) + 150 mg/kg caffeine (MC-150), 20 mg/kg meperidine (2 doses every 6 hours) + 300 mg/kg caffeine (MC-300), 20 mg/kg meperidine for 7 days (2 doses 6 hours apart) +150 mg/kg caffeine (MCU-150), 20 mg/kg meperidine for 7 days (2 doses 6 hours apart) +300 mg/kg caffeine (MCU-300), 7 days 150 mg/kg metyrapone + 20 meperidine (2 doses 6 hours apart) + 150 mg/kg caffeine (MMC- 150) and 7 days 150 mg/kg metyrapone + 20 mg/kg meperidine (2 doses 6 hours apart) + 300 mg/kg caffeine (MMC-300). Their results suggest that normal levels of ADR and cortisol may enhance the analgesic effect and duration of meperidine, as well as prevent epileptic seizures. My comments are listed below-
Comments:
1. What’s the novelty of this research and how have different experiments been designed? This needs to be described clearly. Before the describing the results, briefly describe why different experiments were performed. Authors need to perform some additional experiments to support their claim!
2. There are not much recent research have been added/discussed about the current scenario of meperidine in the present work. What are the possible alternatives of meperidine, is there any molecule already?
3. Figure 1 and Figure 2 doesn’t have error bars on the graph. Add error bars and significance in the graph as well.
4. (*) mark use for different significance in the manuscript not labeled in table 1. Include each significance report for analgesic activity as well! What a (-ve) % of analgesic means, briefly describe in the maintext.
Minor:
1. Section and sub sections throughout the manuscript (Introduction, Materials and method, Results, etc are either 1. or 1.1
2. Font and size of paragraphs are different.
3. Discussion portion is too lengthy, include some in the results sections when describing the result of a particular experiment.
4. Use the correct abbreviation only throughout the text.
Thanks
Author Response
Comments 1) What’s the novelty of this research and how have different experiments been designed? This needs to be described clearly. Before the describing the results, briefly describe why different experiments were performed. Authors need to perform some additional experiments to support their claim!
Response 1: The conclusion part of the summary has been changed. ‘The current study showed that if serum ADR and cortisol levels are kept at normal levels, a longer-lasting and stronger analgesic effect can be achieved with meperidine and epileptic seizures can be prevented.’
Additions were made to the material method section regarding the design of the experiments. Page: 4-7
A healthy group was added to the study.
Comments 2) There are not much recent research have been added/discussed about the current scenario of meperidine in the present work. What are the possible alternatives of meperidine, is there any molecule already?
Response 2: New references about meperidine have been added to the introduction. Page: 3, Line: 89-94
Comments 3) Figure 1 and Figure 2 doesn’t have error bars on the graph. Add error bars and significance in the graph as well.
Response 3: GraphPad Prism 9 program was used for the figures. Error bars could not be added in the chart type used. Figures 1 and 2 were edited in the same program by changing the graphic type. Data are shown as mean+SEM. Significance flags added.
Comments 4) (*) mark use for different significance in the manuscript not labeled in table 1. Include each significance report for analgesic activity as well! What a (-ve) % of analgesic means, briefly describe in the maintext.
Response 4: The description of Table 1 was mistakenly merged with the description of Table 2. The error has been corrected and the description placed below Table 1. Explanation regarding analgesic percentage was added to the material method section. Page: 5, Line: 174-178
Minor:
Comments 1) Section and sub sections throughout the manuscript (Introduction, Materials and method, Results, etc are either 1. or 1.1
Response 1: The text has been arranged formally as suggested.
Comments 2) Font and size of paragraphs are different.
Response 2: The text has been arranged formally as suggested.
Comments 3) Discussion portion is too lengthy, include some in the results sections when describing the result of a particular experiment.
Response 3: The discussion has been reorganized.
Comments 4) Use the correct abbreviation only throughout the text.
Response 4: Abbreviations have been reviewed.

Reviewer 3 Report
The paper submitted for review presents pharmacological study concerning the role of adrenalinÄ™, noradrenaline and cortisol in the pathogenesis of meperidine's analgesic potency, duration and epilepsy-like toxic effect., performed in vivo in animal models. The paper is interesting but needs numerous changes and additions
In the Introduction, the Authors should add more information about the receptors affected by meperidine. Additionally, I suggest extending the statement – „Based on the results of a previous study, it has been found that rats with reduced serum ADR levels experience longer anesthetic effects than healthy rats [10]” - why does this happen?
In the Materials and methods section, the Authors should add information about the numer of animals housed in one cage since it may have a direct impact on the levels of adrenaline and cortisol in these animals.
The weight and age of the animals participating in the experiments should be completed.
Additionally, there is an inaccuracy in the numer of used animals. In the abstract the Authors clearly write that 60 animals took part in the experiment (10 groups of 6 animals each) but in the Animals section only 54. Where do these inaccuracies come from?
In the Chemical substances section, the Authors should describe in detail how the solutions administered to animals were obtained, what was the route of administration and in what volume.
Similarly, the Experimental groups section is described very generally. It is difficult to identify individual groups, the reader must write them down in order to understand the Authors' reasoning, but it is their task! I suggest describing the groups more precisely, including the doses of the administered substances, the route of administration, and the intervals between administration and testing. Please provide a detailed description in points or table.
The Authors should include indications of the statistical significance of results between individual groups on the Figures.
The Authors should add the ANOVA values in the description of results.
Please explain how you managed to obtain statistical significance between groups with only 6 animals? There is high individual variability in behavioral research and it is very difficult to objectively assess the results obtained with such small group sizes.
Additionally, there is no reference substance with a well-known effect, e.g. analgesic, to which the obtained result will be compared. This is crucial for experiments because in behavioral research we are dealing with such a large variability of animal reactions, even depending on environmental conditions, that on different days the reactions may be completely different and the interpretation of the results may be incorrect without checking whether the expected effect is actually achieved.
In my opinion, the entire discussion needs to be reworded. There are frequent repetitions of the same statement, but in fact there is no precise indication of the potential mechanisms that are responsible for the observed effects, and the description of the results appears too often. Additionally, there are stylistic errors that make it much more difficult to understand the Authors' reasoning.
The Authors should explain why: „The animals in the MCU-300 group, which received 300 mg/kg caffeine, all had contractions in an average of 79.33±2.35 seconds, and all of the animals in this group died”.
Stylistic errors should be corrected throughout the work, one font size should be introduced, and individual paragraphs should be numbered consecutively.
Author Response

(The authors gave the same response as above.)

Round 2
Reviewer 2 Report
This revised version looks better to me! Authors have tried to address most of my concerns.
Thanks